# Compute Optimal Inference and Provable Amortisation Gap in Sparse Autoencoders

## Abstract

A recent line of work has shown promise in using sparse autoencoders (SAEs) to uncover interpretable features in neural network representations. However, the simple linear-nonlinear encoding mechanism in SAEs limits their ability to perform accurate sparse inference. In this paper, we investigate sparse inference and learning in SAEs through the lens of sparse coding. Specifically, we show that SAEs perform amortised sparse inference with a computationally restricted encoder and, using compressed sensing theory, we prove that this mapping is inherently insufficient for accurate sparse inference, even in solvable cases. Building on this theory, we empirically explore conditions where more sophisticated sparse inference methods outperform traditional SAE encoders. Our key contribution is the decoupling of the encoding and decoding processes, which allows for a comparison of various sparse encoding strategies. We evaluate these strategies on two dimensions: alignment with true underlying sparse features and correct inference of sparse codes, while also accounting for computational costs during training and inference. Our results reveal that substantial performance gains can be achieved with minimal increases in compute cost. We demonstrate that this generalises to SAEs applied to large language models (LLMs), where advanced encoders achieve similar interpretability. This work opens new avenues for understanding neural network representations and offers important implications for improving the tools we use to analyse the activations of large language models.

## 1 Introduction

Understanding the inner workings of neural networks has become a critical task since these models are increasingly employed in high-stakes decision-making scenarios (Fan et al., 2021; Shahroudnejad, 2021; Räuker et al., 2023). As the complexity and scale of neural networks continue to grow, so does the importance of developing robust methods for interpreting their internal representations. This paper explores the synergy between sparse autoencoders (SAEs) and sparse coding techniques, aiming to advance our ability to extract interpretable features from neural network activations.

Recent work has investigated the "superposition hypothesis" (Elhage et al., 2022), which posits that neural networks represent interpretable features in a linear manner using non-orthogonal directions in their latent spaces. Building on this idea, researchers have shown that individual features can be recovered from these superposed representations using sparse autoencoders (Bricken et al., 2023; Cunningham et al., 2023). These models learn sparse and overcomplete representations of neural activations, with the resulting sparse codes often proving to be more interpretable than the original dense representations (Cunningham et al., 2023; Elhage et al., 2022; Gao et al., 2024).

The mathematical foundation of SAEs aligns closely with that of sparse coding. Both approaches assume that a large number of sparse codes are linearly projected into a lower-dimensional space, forming the neural representation. However, while sparse coding typically involves solving an optimisation problem for each input, SAEs learn an efficient encoding function through gradient descent, potentially sacrificing optimal sparsity for computational efficiency. This trade-off introduces what we term the "amortisation gap" – the disparity between the best sparse code predicted by an SAE encoder and the optimal sparse codes that an unconstrained sparse inference algorithm might produce (Marino et al., 2018).

In this paper, we explore this amortisation gap and investigate whether more sophisticated sparse inference methods can outperform traditional SAE encoders. Our key contribution is decoupling the encoding and decoding processes, allowing for a comparison of various sparse encoding strategies. We evaluate four types of encoding methods on synthetic datasets with known ground-truth features. We evaluate these methods on two dimensions: alignment with true underlying sparse features and inference of the correct sparse codes, while accounting for computational costs during both training and inference. To demonstrate real-world applicability, we also train models on GPT-2 activations (Radford et al., 2019), showing that more complex methods can yield interpretable features in large language models. This approach aims to identify optimal strategies for extracting interpretable features from neural representations across different computational regimes.

## 2 BACKGROUND AND RELATED WORK

### 2.1 SPARSE NEURAL REPRESENTATIONS

Sparse representations in neural networks specifically refer to activation patterns where only a small subset of neurons are active for any given input (Olshausen & Field, 1996). These representations have gained attention due to their potential for improved interpretability and efficiency (Lee et al., 2007). Sparse autoencoders (SAEs) are neural network architectures designed to learn sparse representations of input data (Ng et al., 2011; Makhzani & Frey, 2013). An SAE consists of an encoder that maps input data to a sparse latent space and a decoder that reconstructs the input from this latent representation. Sparse coding, on the other hand, is a technique that aims to represent input data as a sparse linear combination of basis vectors (Olshausen & Field, 1997). The objective of sparse coding is to find both the optimal basis (dictionary) and the sparse coefficients that minimise reconstruction error while maintaining sparsity. While both SAEs and sparse coding seek to find sparse representations, they differ in their approach. SAEs learn an efficient encoding function through gradient descent, allowing for fast inference but potentially sacrificing optimal sparsity. Sparse coding, in contrast, solves an optimisation problem for each input, potentially achieving better sparsity at the cost of increased computational complexity during inference.

### 2.2 SUPERPOSITION IN NEURAL REPRESENTATIONS

The superposition hypothesis suggests that neural networks can represent more features than they have dimensions, particularly when these features are sparse (Elhage et al., 2022). Formally, let us consider a neural representation $y \in \mathbb{R}^M$ and a set of $N$ features, where typically $N > M$. In a linear representation framework, each feature $f_i$ is associated with a direction $w_i \in \mathbb{R}^M$. The presence of multiple features is represented by $y = \sum_{i=1}^{N} x_i w_i$ where $x_i \in \mathbb{R}$ represents the activation or intensity of feature $i$. Features are often defined as interpretable properties of the input that a sufficiently large neural network would reliably dedicate a neuron to representing (Olah et al., 2020).

In an $M$-dimensional vector space, only $M$ orthogonal vectors can fit. However, the Johnson-Lindenstrauss Lemma states that if we permit small deviations from orthogonality, we can fit exponentially more vectors into that space. More formally, for any set of $N$ points in a high-dimensional space, there exists a linear map to a lower-dimensional space of $\mathcal{O}(\log N / \epsilon^2)$ dimensions that preserves pairwise distances up to a factor of $(1 \pm \epsilon)$. This lemma supports the hypothesis that LLMs might be leveraging a similar principle in superposition.

Superposition occurs when the matrix $W = [w_1, ..., w_N] \in \mathbb{R}^{M \times N}$ has more columns than rows (i.e., $N > M$), making $W^T W$ non-invertible. Superposition relies on the sparsity of feature activations. Let $s = |x|_0$ be the number of non-zero elements in $x = [x_1, ..., x_N]^T$. When $s \ll N$, the model can tolerate some level of interference between features, as the probability of many features being active simultaneously (and thus interfering) is low.

### 2.3 COMPRESSED SENSING AND SPARSE CODING

Compressed sensing theory provides a framework for understanding how sparse signals can be recovered from lower-dimensional measurements (Donoho, 2006). This theory suggests that under

certain conditions, we can perfectly recover a sparse signal from fewer measurements than traditionally required by the Nyquist-Shannon sampling theorem. Let $s \in \mathbb{R}^N$ be a sparse signal with at most $K$ non-zero components. If we make $M$ linear measurements of this signal, represented as $y = Ws$ where $W \in \mathbb{R}^{M \times N}$, compressed sensing theory states that we can recover $s$ from $y$ with high probability if:

$$M > \mathcal{O}\left(K \log\left(\frac{N}{K}\right)\right) \tag{1}$$

This result holds under certain assumptions about the measurement matrix $W$, such as the Restricted Isometry Property (RIP) (Candes, 2008). Sparse coding is one approach to recovering such sparse representations. The objective function for sparse coding (Olshausen & Field, 1996) is:

$$\mathcal{L}(D, \alpha) := \sum_i^n |x_i - D\alpha_i|_2^2 + \lambda |\alpha_i|_1 \tag{2}$$

where $D \in \mathbb{R}^{k \times m}$ is the dictionary, $\alpha_i \in \mathbb{R}^m$ are the sparse codes for data point $x_i \in \mathbb{R}^k$, and $\lambda$ is a hyperparameter controlling sparsity. Optimisation of this objective typically alternates between two steps. First is sparse inference: $\min_{\alpha} \sum_i^n |x_i - D\alpha_i|_2^2 + \lambda |\alpha_i|_1$. Then dictionary learning: $\min_{D} \sum_i^n |x_i - D\alpha_i|_2^2$ s.t. $\forall i \in 1, ..., m : |D:, i| = 1$. These techniques allow extraction of interpretable, sparse representations from high-dimensional neural data.

## 2.4 Sparse Autoencoders

Sparse autoencoders (SAEs) offer an alternative approach to extracting sparse representations, using amortised inference instead of the iterative optimisation used in sparse coding. SAEs learn to reconstruct inputs using a sparse set of features in a higher-dimensional space, potentially disentangling superposed features (Elhage et al., 2022; Olshausen & Field, 1997). The architecture of an SAE consists of an encoder network that maps the input to a hidden, sparse representation of latent coefficients, and a decoder network that reconstructs the input as a linear combination of vectors, with the coefficients defined by the sparse representation. Let $x_i \in \mathbb{R}^k$ be an input vector (as in our sparse coding formulation), and $\alpha_i \in \mathbb{R}^m$ be the hidden representation (analogous to the sparse codes in sparse coding), where typically $m > k$. The encoder and decoder functions are defined as:

$$\text{Encoder}: \quad \alpha_i = f_\theta(x_i) = \sigma(W_e x_i + b_e) \tag{3}$$
$$\text{Decoder}: \quad \hat{x}_i = g_\phi(\alpha_i) = W_d \alpha_i + b_d \tag{4}$$

where $W_e \in \mathbb{R}^{m \times k}$ and $W_d \in \mathbb{R}^{k \times m}$ are the encoding and decoding weight matrices, $b_e \in \mathbb{R}^m$ and $b_d \in \mathbb{R}^k$ are bias vectors, and $\sigma(\cdot)$ is a non-linear activation function (e.g., ReLU). The parameters $\theta = W_e, b_e$ and $\phi = W_d, b_d$ are learned during training. The training objective of an SAE combines reconstruction loss with a sparsity constraint:

$$\mathcal{L}(\theta, \phi) = \frac{1}{n} \sum_{i=1}^n |x_i - \hat{x}_i|_2^2 + \lambda \mathcal{L}_{\text{sparse}}(\alpha_i) \tag{5}$$

where $\lambda > 0$ is a hyperparameter controlling the trade-off between reconstruction fidelity and sparsity. The sparsity loss $\mathcal{L}_{\text{sparse}}(\alpha_i)$ is often L1 regularisation $\mathcal{L}_{\text{sparse}}(\alpha) = ||\alpha||_1$.

Comparing this formulation to sparse coding, we can see that SAEs provide an amortised inference method by learning an encoder function $f_\theta$ that directly maps inputs to sparse codes. This contrasts with the iterative optimisation process used in sparse coding for inference.

**SAE with Inference-Time Optimisation (SAE+ITO)** (SAE+ITO) is an extension of the standard SAE approach that combines the learned dictionary from SAEs with inference-time optimisation for sparse code inference (Nanda et al., 2024). In this method, the decoder weights $W_d$ learned during SAE training are retained, but the encoder function $f_\theta$ is replaced with an optimisation procedure at inference time. For each input $x_i$, SAE+ITO solves the following optimisation problem:

$$\alpha_i^* = \arg\min_{\alpha_i} |x_i - W_d \alpha_i|_2^2 + \lambda |\alpha_i|_1 \tag{6}$$

where $\lambda$ controls the sparsity of the solution. This formulation allows for potentially more accurate sparse codes by directly minimising reconstruction error, rather than relying on the learned

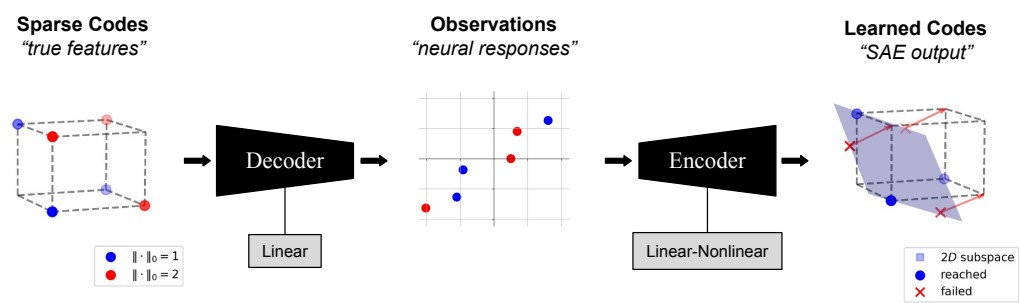

Figure 1: **Illustration of SAE Amortisation Gap. Left**, shows sparse sources in an $N = 3$ dimensional space with at most $\|s\| \leq K = 2$ non-zero entries. Both blue and red points are valid sources, by contrast, the top right corner $s = (1, 1, 1)$ is not. **Middle**, shows the sources as they are linearly *decoded* into observation space. This is, in most applications, the activation space of a neural network that we are trying to lift out of superposition. **Right**, shows how using a linear-nonlinear encoder, a SAE is tasked to project the points back onto their correct positions. This is not possible, because the pre-activations are at most $M = 2$ dimensional (see proof in Appendix A).

encoder approximation. While this approach incurs higher computational costs during inference, it can potentially achieve better reconstruction quality and more flexible control over sparsity levels without retraining the entire model. The optimisation problem can be solved using algorithms such as matched pursuit (Blumensath & Davies, 2008) and gradient pursuit (Nanda et al., 2024).

## 2.5 APPLICATIONS IN NEURAL NETWORK MODELS

Sparse autoencoders (SAEs) have emerged as a promising tool for enhancing the interpretability of large language models (LLMs) by extracting interpretable features from their dense representations. Early work by Cunningham et al. (2023) and Bricken et al. (2023) demonstrated the potential of sparse dictionary learning to untangle features, lifting them out of superposition in transformer MLPs. This approach was extended to attention heads by Kissane et al. (2024), who scaled it to GPT-2 (Radford et al., 2019). These studies have shown that SAEs can extract highly abstract, multilingual, and multimodal features from LLMs, including potentially safety-relevant features related to deception, bias, and dangerous content (Templeton, 2024). In vision models, Gorton (2024) and Klindt et al. (2023) trained SAEs on convolutional neural network activations. The latter found that K-means (which is equivalent to one-hot sparse coding) outperformed SAEs (Fig.12) in quantitative interpretability metrics (Zimmermann et al., 2024).

The scaling of SAEs to larger models has been a focus of recent research, with significant progress made in applying them to state-of-the-art LLMs. Gao et al. (2024) proposed using k-sparse autoencoders (Makhzani & Frey, 2013) to simplify tuning and improve the reconstruction-sparsity frontier, demonstrating clean scaling laws with respect to autoencoder size and sparsity. They successfully trained a 16 million latent autoencoder on GPT-4 activations. Similarly, Templeton (2024) reported extracting high-quality features from Claude 3 Sonnet, while Lieberum et al. (2024) released a comprehensive suite of SAEs trained on all layers of Gemma 2 models. These advancements underscore the importance of developing efficient and accurate SAE techniques to reduce the amortisation gap, especially as applications to larger models become more prevalent. The growing body of work on SAEs in LLMs suggests that they may play a crucial role in future interpretability research.

## 3 METHODS

This section outlines our approach to comparing sparse encoding strategies. We begin by presenting a theoretical foundation for the suboptimality of sparse autoencoders (SAEs), followed by our data generation process, encoding schemes, evaluation metrics, and experimental scenarios.

## 3.1 THEORY: PROVABLE SUBOPTIMALITY OF SAEs

**Theorem 1** (SAE Amortisation Gap). *Let $S = \mathbb{R}^N$ be $N$ sources following a sparse distribution $P_S$ such that any sample has at most $K \geq 2$ non-zero entries, i.e., $\|s\|_0 \leq K, \forall s \in supp(P_S)$. The sources are linearly projected into an $M$-dimensional space, satisfying the restricted isometry property, where $K \log \frac{N}{K} \leq M < N$. A sparse autoencoder (SAE) with a linear-nonlinear (L-NL) encoder must have a non-zero amortisation gap.*

The complete proof of Theorem 1 is provided in Appendix A. The theorem considers a setting where sparse signals $s \in \mathbb{R}^N$ with at most $K$ non-zero entries are projected into an $M$-dimensional space ($M < N$). Compressed sensing theory guarantees that unique recovery of these sparse signals is possible when $M \geq K \log(N/K)$, up to sign ambiguities (Donoho, 2006). However, we prove that SAEs fail to achieve this optimal recovery, resulting in a non-zero amortisation gap. The core of this limitation lies in the architectural constraints of the SAE's encoder. The linear-nonlinear (L-NL) structure of the encoder lacks the computational complexity required to fully recover the high-dimensional ($N$) sparse representation from its lower-dimensional ($M$) projection. Figure 1 illustrates this concept geometrically.

## 3.2 SYNTHETIC DATA

To evaluate our sparse encoding strategies, we generate synthetic datasets with known ground-truth latent representations and dictionary vectors. We first construct a dictionary matrix $\mathbf{D} \in \mathbb{R}^{M \times N}$, where each column represents a dictionary element. We then generate latent representations $\mathbf{s}_i \in \mathbb{R}^N$ with exactly $K$ non-zero entries ($K \ll N$), drawn from a standard normal distribution. This allows us to create observed data points as $\mathbf{x}_i = \mathbf{D}\mathbf{s}_i + \boldsymbol{\epsilon}_i$, where $\boldsymbol{\epsilon}_i \sim \mathcal{N}(0, \sigma^2 \mathbf{I})$ represents additive Gaussian noise. This process yields a dataset $\mathcal{D} = (\mathbf{x}_i, \mathbf{s}_i)_{i=1}^n$, where $\mathbf{x}_i \in \mathbb{R}^M$ and $\mathbf{s}_i \in \mathbb{R}^N$.

## 3.3 SPARSE ENCODING SCHEMES

We compare four sparse encoding strategies:

1. **Sparse Autoencoder (SAE)**: $f(x) := \sigma(Wx)$, where $\sigma$ is a nonlinear activation function.

2. **Multilayer Perceptron (MLP)**: $f(x) := \sigma(W_n \sigma(W_{n-1} \ldots \sigma(W_1 x)))$, with the same decoder as the SAE.

3. **Sparse Coding (SC)**: $f(x) = \arg\min_{\hat{s}} |x - D\hat{s}|_2^2 + \lambda ||\hat{s}||_1$, solved iteratively with $s_{t+1} = s_t + \eta \nabla \mathcal{L}$, where $\mathcal{L}$ is the MSE loss with L1 penalty.

4. **SAE with Inference-Time Optimisation (SAE+ITO)**: Uses the learned SAE dictionary, optimising sparse coefficients at inference time.

For all methods, we normalise the column vectors of the decoder matrix to have unit norm, preventing the decoder from reducing the sparsity loss $||\hat{s}||_1$ by increasing feature vector magnitudes.

## 3.4 MEASURING THE QUALITY OF THE ENCODER AND DECODER

For any given $x$, how do we measure the quality of (1) the encoding (i.e. the sparse coefficients); and (2) the decoding (i.e. the actual reconstruction, given the coefficients)?

We employ the Mean Correlation Coefficient (MCC) to evaluate both encoder and dictionary quality:

$$\text{MCC} = \frac{1}{d} \sum_{(i,j) \in M} |c_{ij}| \tag{7}$$

where $c_{ij}$ is the Pearson correlation coefficient between the $i$-th true feature and the $j$-th learned feature, and $M$ is the set of matched pairs determined by the Hungarian algorithm (or a greedy approximation when dimensions differ). This metric quantifies alignment between learned sparse coefficients and true underlying sparse features (encoder quality), and learned dictionary vectors and true dictionary vectors (dictionary quality).

## 3.5 Disentangling Dictionary Learning and Sparse Inference

Our study decomposes the sparse coding problem into two interrelated tasks: dictionary learning and sparse inference. Dictionary learning involves finding an appropriate sparse dictionary $D \in \mathbb{R}^{M \times N}$ from data, while sparse inference focuses on reconstructing a signal $x \in \mathbb{R}^M$ using a sparse combination of dictionary elements, solving for $s \in \mathbb{R}^N$ in $x \approx Ds$ where $s$ is sparse. These tasks are intrinsically linked: dictionary learning often involves sparse inference in its inner loop, while sparse inference requires a dictionary.

**Known Sparse Codes.** In this scenario, we assume knowledge of the true sparse codes $s^*$ and focus solely on the encoder's ability to predict these latents, effectively reducing the problem to latent regression. We define the objective as minimising $\mathcal{L}(f(x), s^*) = 1 - \cos(f(x), s^*)$, where $f$ is the encoding function and $\cos$ denotes cosine similarity.[1] In this setting, only the SAE encoder and MLP are applicable, as they directly learn mappings from input to latent space. The SAE encoder learns an amortised inference function, while the MLP learns a similar but more complex mapping. Conversely, SAE+ITO and sparse coding are not suitable for this task. SAE+ITO focuses on optimising reconstruction using a fixed dictionary, which is irrelevant when true latents are known. Similarly, sparse coding alternates between latent and dictionary optimisation, which reduces to encoder training when the dictionary is disregarded.

**Known Dictionary.** When the true dictionary $D^*$ is known, we focus on optimising the encoder or inference process while keeping the dictionary fixed. This scenario is applicable to SAE, MLP, and SAE+ITO methods. For SAE and MLP, we optimise $\min_\theta \mathbb{E}_x[\|x - D^* f_\theta(x)\|_2^2]$, where $f_\theta$ represents the encoder function with parameters $\theta$. SAE+ITO, in contrast, performs gradient-based optimisation at inference time: $\min_s |x - D^* s|_2^2 + \lambda |z|_1$ for each input $x$, incurring zero training FLOPs but higher inference-time costs. This differs from SAE and MLP by directly optimising latent coefficients rather than learning an encoding function. Sparse coding is not applicable in this scenario, as it reduces to SAE+ITO when the dictionary is known and fixed.

**Unknown Sparse Codes and Dictionary.** This scenario represents the standard setup in sparse coding, where both the sparse codes $s$ and the dictionary $D$ are unknown and must be learned simultaneously. All four methods—SAE, MLP, SAE+ITO, and sparse coding—are applicable in this context, each approaching the problem differently. SAE and MLP learn both an encoder function $f_\theta(x)$ and a dictionary $D$ simultaneously. SAE+ITO and sparse coding learn a dictionary during training and optimises latents at inference time.

## 4 Synthetic Sparse Inference Experiments

We present the results of our experiments comparing different sparse encoding strategies across various scenarios. To provide a minimal setting for investigating the phenomena of interest, all experiments were conducted using synthetic data with $N = 16$ sparse sources, $M = 8$ measurements, and $K = 3$ active components per timestep, unless otherwise specified (more settings in Sec. 4.4).

### 4.1 Known Sparse Codes

We first compare the performance of sparse autoencoders (SAEs) and multilayer perceptrons (MLPs) in predicting known latent representations. Figure 2 illustrates the performance of SAEs and MLPs with varying hidden layer widths. MLPs consistently outperform SAEs in terms of Mean Correlation Coefficient (MCC), with wider hidden layers achieving higher performance (Figure 2a). The MLP with $H = 1024$ reaches an MCC approximately 0.1 higher than the SAE at convergence. While MLPs converge faster in terms of training steps, this comes at the cost of increased computational complexity (Figure 2b). All MLPs surpass the SAE's plateau performance at approximately the same total FLOPs, suggesting a consistent computational threshold beyond which MLPs become more effective, regardless of hidden layer width.

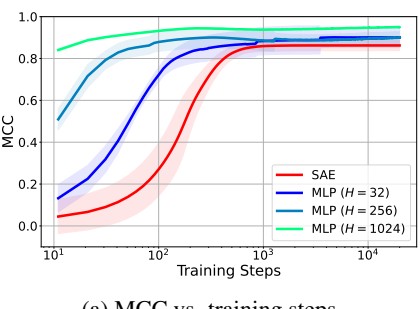 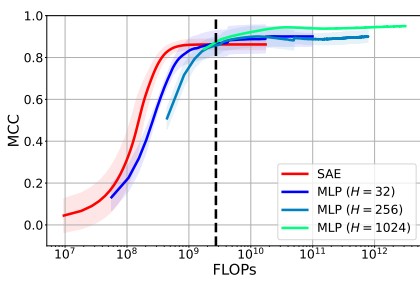

(a) MCC vs. training steps       (b) MCC vs. total FLOPs

Figure 2: Performance comparison of SAE and MLPs in predicting known latent representations. The black dashed line in (b) indicates the average FLOPs at which MLPs surpass SAE performance.

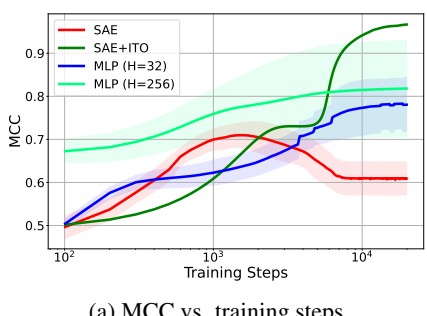 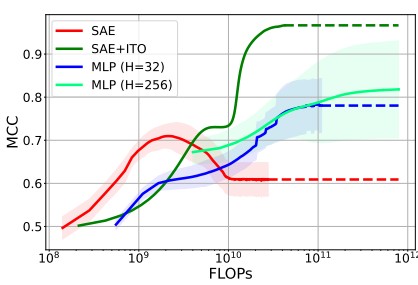

(a) MCC vs. training steps       (b) MCC vs. total FLOPs

Figure 3: Performance comparison of SAE, SAE with inference-time optimisation (SAE+ITO), and MLPs in predicting latent representations with a known dictionary. Dashed lines in (b) indicate extrapolated performance beyond the measured range.

## 4.2 KNOWN DICTIONARY

Next, we examine the performance of different encoding strategies when the true dictionary $D^*$ is known. Figure 3 shows the performance of SAE, SAE+ITO, and MLPs. MLPs consistently outperform the standard SAE, achieving an MCC nearly $10\%$ higher at convergence (Figure 3a). Both MLP configurations ($H = 32$ and $H = 256$) converge to similar performance levels, with the wider network showing faster initial progress. When plotted against total FLOPs, the MLP curves overlap, suggesting a consistent computational cost-to-performance ratio across different hidden layer widths (Figure 3b). SAE+ITO initialised with SAE latents exhibits distinct, stepwise improvements throughout training, ultimately achieving the highest MCC.

## 4.3 UNKNOWN SPARSE CODES AND DICTIONARY

Finally, we evaluate all four methods when both latent representations and dictionary are unknown. We use a dataset of $2048$ samples, evenly split between training and testing sets, and conduct 5 independent runs of $100,000$ training steps each.

Figures 4 illustrates the performance in latent prediction and dictionary learning, respectively. For latent prediction, SAE, SAE+ITO, and MLPs converge to similar MCC, with MLPs showing a slight advantage. Sparse coding demonstrates superior performance, achieving an MCC over $10\%$ higher than other methods, despite an initial decrease in performance. Sparse coding reaches this higher performance while using comparable FLOPs to the MLP with $H = 256$. For dictionary learning, both MLPs and sparse coding outperform SAE by a margin of approximately $10\%$. Sparse coding again exhibits an initial decrease in dictionary MCC before surpassing other methods.

---

[1] We use cosine similarity rather than MSE loss in this setting because we found training to be more stable.

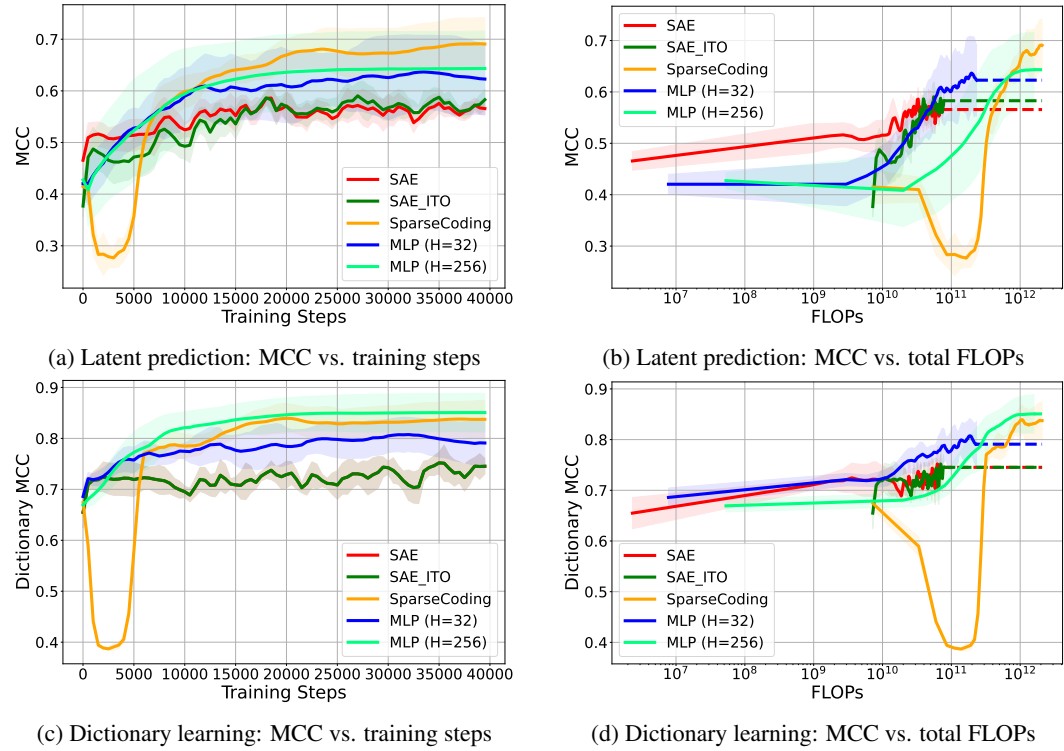

(a) Latent prediction: MCC vs. training steps      (b) Latent prediction: MCC vs. total FLOPs

(c) Dictionary learning: MCC vs. training steps      (d) Dictionary learning: MCC vs. total FLOPs

Figure 4: Dictionary learning performance comparison when both $s^*$ and $D^*$ are unknown.

### 4.4 PERFORMANCE ACROSS VARYING DATA REGIMES

To understand how performance varies with changes in data characteristics, we trained models under varying $N$, $M$, and $K$, holding other hyperparameters constant.

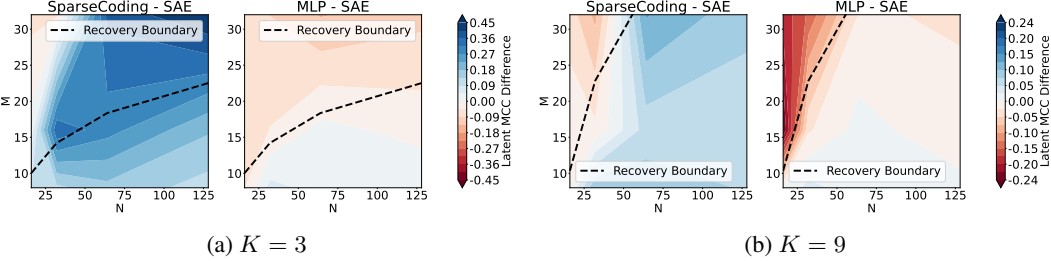

(a) $K = 3$            (b) $K = 9$

Figure 5: Difference in final latent MCC between methods across varying $N$ and $M$, for $K = 3$ and $K = 9$. **Left:** Sparse coding vs. SAE. **Right:** MLP vs. SAE. The black dashed line indicates the theoretical recovery boundary.

Figure 5 shows the difference in final latent MCC between methods. Sparse coding outperforms SAE in essentially all data-generation regimes, for both $K = 3$ and $K = 9$. MLP and SAE perform roughly equivalently, with MLP slightly better as $M$ (number of measurements) increases. The performance advantage of sparse coding is more pronounced in regimes where compressed sensing theory predicts recoverability (above and to the left of the black dashed line).

**Sparsity-Performance Trade-off**   We also investigated the trade-off between sparsity and performance for each method in Figure 6. Sparse coding achieves slightly lower reconstruction error for each L0 level, barring some very small active latents. Sparse coding shows a Pareto improvement at each L0 level in terms of MCC, even with very small active latents. The improvement is more evident when plotting against L1 rather than L0, as L1 accounts for the magnitude of non-zero val-

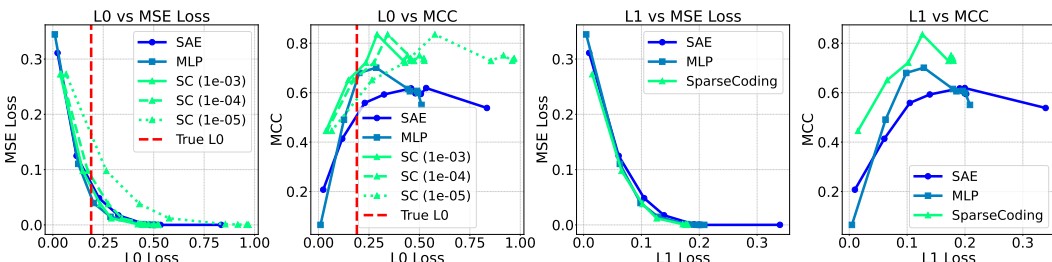

Figure 6: Pareto curves showing sparsity (L0 or L1 loss) against performance (MSE loss or latent MCC) for models trained with varying L1 penalty coefficients $\lambda$. The red dashed line in the top row shows the true L0 of the sparse sources. Multiple thresholds for active features are shown for sparse coding due to the presence of very small non-zero values.

ues. The presence of very small non-zero latents in sparse coding motivates the exploration of top-$k$ sparse coding, detailed in Appendix F.2.

## 5 INTERPRETABILITY OF SPARSE CODING SCHEMES

A common criticism of more powerful encoding techniques is that they can find concepts that are more difficult to interpret, or that are not actually used by the model. To investigate the interpretability of more complex encoding techniques, we trained both an SAE and MLP on 406 million tokens from OpenWebText. The MLP used a hidden width of 4224 and both models had 16,896 sparse codes. The models were trained on the residual-stream pre-activations of Layer 9 in GPT-2 Small, which have a dimension of 768. We used a learning rate of $3 \cdot 10^{-4}$ and an $L_1$ penalty of $1 \cdot 10^{-4}$. Throughout training, we tracked normalised mean squared error (MSE), dividing it by the error when predicting the mean activations, as well as $L_0$ sparsity. The final SAE achieved a normalised MSE of 0.062 with an average $L_0$ of 39.55, while the MLP reached a normalised MSE of 0.051 with an average $L_0$ of 40.20. The final models had a significant number of dead neurons (measured as not having fired in the last 50,000 training steps): 71% for the SAE and 66% for the MLP.

To assess the interpretability of the learned features, we selected 200 random features that activated on our test set of 13.1 million OpenWebText tokens. For each feature, we employed an automated interpretability classification approach. We identified the top 10 activating examples for the feature in our test dataset and labelled the token with the highest activation. We also computed logit effects for each feature through the path expansion $W_U \cdot f$, where $W_U$ is the model's unembedding matrix and $f$ is the feature vector. The top 10 and bottom 10 tokens resulting from this logit effect calculation were noted.

We provided both the activating examples and the top and bottom promoted logits to GPT-4, which was instructed to construct a precise explanation of the feature's function (prompt in Appendix H). To evaluate the accuracy of these interpretations, we presented them to another instance of GPT-4 along with 5 new activating examples and 5 non-activating examples, labeling the token on which the feature potentially activates. The model was then asked to predict which of these examples the feature would activate on, and we calculated the F1-score compared to the ground truth.

Figure 7 displays the distributions of the F1 scores for the 200 SAE and MLP characteristics. The results indicate that the MLP features demonstrate interpretability comparable to the SAE features.

## 6 DISCUSSION

Our study presents both theoretical and empirical evidence for the existence of an inherent amortisation gap in sparse autoencoders (SAEs) when applied to neural network interpretability tasks. We prove that SAEs with linear-nonlinear encoders cannot achieve optimal sparse recovery in settings where such recovery is theoretically possible. This limitation is corroborated by our experimental results, which demonstrate superior performance of more complex encoding methods, such as multilayer perceptrons (MLPs) and sparse coding, across various synthetic data scenarios. Notably, our

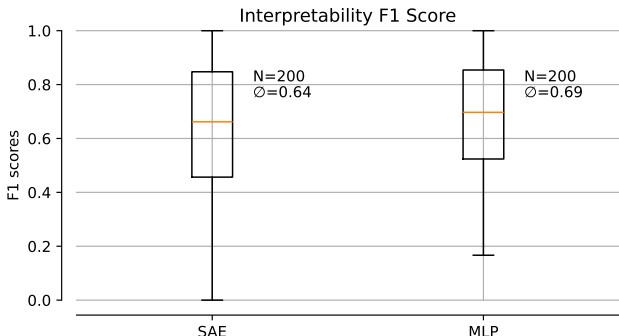

Figure 7: Distribution of F1 scores for feature interpretability of an SAE and an MLP trained on residual stream activations of Layer 9 in GPT-2.

investigation of GPT-2 activations reveals that MLP-based features exhibit interpretability compara-ble to, and in some cases exceeding, that of SAE features. These findings challenge the prevailing assumption that simpler encoders are necessary for maintaining interpretability. The results of our study have significant implications for the field of neural network interpretability, particularly in the context of large language models. They suggest that more sophisticated encoding techniques can potentially improve feature extraction and interpretability without compromising the validity of the extracted features. However, this potential improvement comes with increased computational costs. Our work provides a framework for exploring this trade-off.

The use of simple linear-nonlinear encoders in SAEs for language model interpretability has been primarily motivated by concerns that more powerful methods might extract features not actually utilised by the transformer (Bricken et al., 2023). However, this approach may be overly conservative given the complexity of representations in transformer layers, which result from multiple rounds of attention and feed-forward computations. Better encoders aligns with recent work on inference-time optimisation (Nanda et al., 2024), and will be validated as we improve encoding evaluation (Makelov et al., 2024). Regardless, SAEs are sensitive to hyperparameters and fragile (Cunningham et al., 2023), so exploring more powerful encoders is warranted.

The computational cost of more complex encoders should be weighed against potential benefits in feature extraction and interpretability. Given the significant resources already invested in projects like Gemma Scope (Lieberum et al., 2024), allocating additional compute to enhance representation quality before decoding may be justified. Importantly, more sophisticated encoders can still maintain the linear decoder necessary for downstream tasks such as steering. Empirical studies comparing feature quality across encoder complexities will be important, as will addressing concerns about non-zero-centered representations raised by Hobbhahn (2023).

**Limitations** Our study has several limitations that warrant consideration. We primarily explored scenarios with constant sparsity and uncorrelated channels in the sparse representation, which may not fully capture the complexity of real-world data. Additionally, our analysis focused on vanilla implementations of the models, which are susceptible to issues like shrinkage due to the L1 penalty. To comprehensively understand the current amortisation gap, future work should incorporate recent SAE variants such as top-k SAEs (Makhzani & Frey, 2013; Gao et al., 2024) and JumpReLU SAEs (Rajamanoharan et al., 2024b), as discussed in Appendix F.1. This extension would allow us to anal-yse how large the amortisation gap is with the new SAE architectures. Similarly, our implementation of SAE+ITO did not use advanced techniques like matched pursuit or gradient pursuit, potentially underestimating its performance. The traditional dictionary learning approaches explored in Ap-pendix F suggest room for improvement in this area. Lastly, our synthetic data generation process did not account for varying feature importance, a key aspect of Elhage et al. (2022)'s framework. Addressing these limitations in future research would provide a more comprehensive understanding of sparse encoding strategies and their applicability to complex neural representations.

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

CONTENTS

## A  AMORTISATION GAP PROOF

**Theorem 1** (SAE Amortisation Gap). *Let $S = \mathbb{R}^N$ be $N$ sources following a sparse distribution $P_S$ such that any sample has at most $K \geq 2$ non-zero entries, i.e., $\|s\|_0 \leq K, \forall s \in supp(P_S)$. The sources are linearly projected into an $M$-dimensional space, satisfying the restricted isometry property, where $K \log \frac{N}{K} \leq M < N$. A sparse autoencoder (SAE) with a linear-nonlinear (L-NL) encoder must have a non-zero amortisation gap.*

This setting is solvable according to compressed sensing theory Donoho (2006), meaning that it is possible to uniquely recover the true $S$ up to sign flips – we cannot resolve the ambiguity between the sign of any code element and the corresponding row in the decoding matrix. If a SAE fails to achieve the same recovery, then there must be a non-zero amortisation gap, meaning that the SAE cannot solve the sparse inference problem of recovering all sparse sources from their $M$-dimensional projection. The problem is the low computational complexity of the L-NL encoder as we see by looking at its functional mapping. Essentially, the SAE is not able, not even after the nonlinear activation function, to recover the high dimensionality ($N$) of the data after a projection into a lower ($M$) dimensional space Figure 1.

*Proof.* Let $S = \text{diag}(s_{11}, ..., s_{NN})$ be a diagonal matrix with non-zero diagonal elements $s_{ii} \neq 0, \forall i \in \{1, ..., N\}$. Ever row $s_i$ is a valid source signal because it has non-zero support under $P_S$ since, $\|s_i\|_0 = 1 \leq K, \forall i \in \{1, ..., N\}$. Let $W_d \in \mathbb{R}^{N \times M}$ be the unknown projection matrix from $N$ down to $M$ dimensions and $W_e \in \mathbb{R}^{M \times N}$ be the learned encoding matrix of the SAE. Define $W := W_d W_e \in \mathbb{R}^{N \times N}$ and

$$S' := SW \tag{8}$$

the pre-activation matrix from the encoder of the SAE. Since $W_d$ projects down into $M$ dimensions,

$$\text{rank}(W) = \text{rank}(W_d W_e) \leq M. \tag{9}$$

It follows that

$$\text{rank}(S') = \text{rank}(SW) \leq M. \tag{10}$$

As an intermediate results, we conclude that the pre-activations $S'$ of the SAE encoder cannot recover the sources $S' \neq |S|$ since $\text{rank}(|S|) = N$, because $S$ is a diagonal matrix.

The next step is to see whether the nonlinear activation function might help to map back to the sources. The SAE must learn an encoding matrix $W_e$ such that

$$|S| = \max(0, SW_d W_e) = \max(0, SW) = \max(0, S') \tag{11}$$

where $\max(0, \cdot)$ is the ReLU activation function. Thus, for the SAE to correctly reconstruct the sparse signals up to sign flips, for any source code $\sigma \in \text{supp}(P_S)$, we require

$$(\sigma W)_i = \begin{cases} |\sigma_i| & \text{if } \sigma_i \neq 0 \\ \leq 0 & \text{otherwise} \end{cases} \tag{12}$$

specifically, $S'$ must be non-positive off the diagonal and identical to $|S|$ on the diagonal.

**Approach:** Show that a matrix $S'$ cannot simultaneously satisfy conditions (eq. 10) and (eq. 12).

According to (eq. 8) and condition (eq. 12), we require that

$$s_1 W = (s'_{11}, s'_{12}, s'_{13}, ..., s'_{1N}) = (|s_{11}|, s'_{12}, s'_{13}, ..., s'_{1N}) \tag{13}$$

with $s'_{1i} \leq 0$ for all $i \in \{2, ..., N\}$. Analogously,

$$s_2 W = (s'_{21}, s'_{22}, s'_{23}, ..., s'_{2N}) = (s'_{21}, |s_{22}|, s'_{23}, ..., s'_{2N}) \tag{14}$$

with $s'_{2i} \leq 0$ for all $i \in \{1, 3, ..., N\}$. Moreover, since $\|s_1 + s_2\|_0 = 2 < K$ we know that $s_1 + s_2$ has non-zero support under $P_S$, so condition (eq. 12) must also hold for it. Thus, we need that

$$\begin{aligned}(s_1 + s_2)W &= (|s_{11} + s_{21}|, |s_{12} + s_{22}|, \gamma_1, ..., \gamma_{N-2}) \\ &= (|s_{11} + 0|, |0 + s_{22}|, \gamma_1, ..., \gamma_{N-2}) \\ &= (|s_{11}|, |s_{22}|, \gamma_1, ..., \gamma_{N-2})\end{aligned} \tag{15}$$

with some non-positive $\gamma_i \leq 0$ for all $i \in \{1, ..., N-2\}$. However, because of linearity,

$$\begin{aligned}(|s_{11}|, |s_{22}|, \gamma_1, ..., \gamma_{N-2}) &= (s_1 + s_2)W \\ &= s_1 W + s_2 W \\ &= (|s_{11}|, s'_{12}, s'_{13}, ..., s'_{1N}) + (s'_{21}, |s_{22}|, s'_{23}, ..., s'_{2N}) \\ &= (|s_{11}| + s'_{21}, s'_{12} + |s_{22}|, s'_{13} + s'_{23}, ..., s'_{1N} + s'_{2N})\end{aligned} \tag{16}$$

Thus, $|s_{11}| = |s_{11}| + s'_{21}$ and $|s_{22}| = s'_{12} + |s_{22}|$. From which it follows that $s'_{21} = 0$ and $s'_{12} = 0$. By repeating this for all $s_i, s_j$ combinations, we obtain that all off-diagonal elements in $S'$ must be zero. However, that means $S' = \text{diag}(|s_{11}|, ..., |s_{NN}|)$ must be diagonal. This leads to a contradiction, since it would imply that $\text{rank}(S') = N$, violating condition (eq. 10). $\square$

**Notes:** We can generalise the result to any sparse distribution $P_S$ with samples $\|s\|_1 \leq k$ for some $k > 0$. In this case, we would choose $\|s_1\| < \frac{k}{2}$ and $\|s_2\| < \frac{k}{2}$. Thus, again we would have $(s_1 + s_2) \in \text{supp}(P_S)$ since $\|s_1 + s_2\| < k$, allowing the same reasoning.

## B  CONCEPTUAL MODELLING FRAMEWORK

The concept of distributed representations in neural networks originated with the Parallel Distributed Processing (PDP) movement (Rumelhart et al., 1986). This work explored how information could be encoded across multiple units in a network, rather than in localised, symbolic representations (Thorpe, 1989). A distributed representation of information means that no single processing unit in a network performs a syntactically or semantically determinable subtask alone. Instead, an assembly of processing units generates a "distributed pattern of activation" to represent information (Smolensky et al., 2006; Van Gelder, 1990; Gelder, 1992).

We show in Figure 8 the general modelling framework we are studying for uncovering these distributed representations. Inputs pass through a neural network and generate some internal neural representation, that is often distributed and in superposition. We use an encoding process to determine active latents, or features, inherent in that process, and a learned decoder to specify what those feature directions should be. This lies at the heart of all methods studied: inference-based methods (i.e., sparse coding), amortised methods (i.e., sparse autoencoders and autoencoders with a more powerful encoder, such as an MLP) and hybrid approaches (i.e., sparse autoencoder with inference-time optimisation).

## C  DECODER WEIGHT ANALYSIS

A useful method for gaining insight into the behavior of our models is through examining the final weights of the decoder. Specifically, we visualize $W^\top W$, an $N \times N$ matrix, for three scenarios: when $N$ equals the true sparse dimensionality, when $N$ exceeds it, and when $N$ is smaller than the true dimensionality.

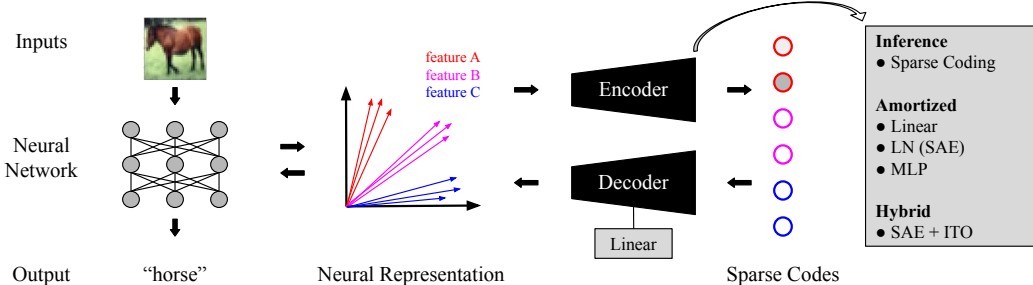

Figure 8: **Illustration of Modelling Framework.** Representations in neural networks (left), commonly represent input features in superposition (center left) (Elhage et al., 2022). Autoencoders can be used to extract sparse (interpretable) codes from neural representations (center right). While the decoder is fixed to be linear (an important assumption), the encoder can be more flexible. Different options for the encoder include *inference*, *amortised* inference and *hybrid* version of both (ITO, inference-time optimisation) (right). Moreover, the encoder might be distinct between training and testing time.

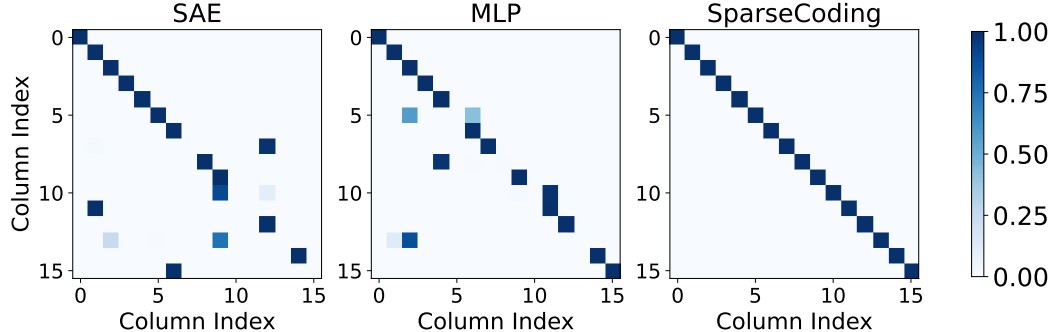

Figure 9: Visualisation of $D^\top D$ when $N$ matches the true sparse dimension. Sparse coding achieves near-identity matrices, while sparse autoencoders (SAE) and multilayer perceptrons (MLP) show significant off-diagonal elements, indicating superposition.

In the case where $N$ matches the true sparse dimension, we observe the matrix $D^\top D$ for the learned decoder matrix $D$ after training. Figure 9 illustrates this scenario for $N = 16$ and $M = 8$, without applying decoder column unit normalisation. For sparse coding, the matrix $D^\top D$ is approximately an $N \times N$ identity matrix after softmax normalisation. This means that the model has learned a set of basis vectors where each column of $D$ is nearly orthogonal to all others, indicating that the features are independent.

In contrast, both the sparse autoencoder (SAE) and the multilayer perceptron (MLP) show $D^\top D$ matrices with a mix of diagonal and off-diagonal elements. In these cases, many off-diagonal elements are close to 1.0, suggesting that these models utilise superposition, representing more features than there are dimensions. This is suboptimal in this particular scenario because the models have the exact number of dimensions required to represent the feature space effectively. Notably, this superposition effect diminishes when vector normalisation is applied during training.

We observe similar patterns when $N$ is greater than the true sparse dimensionality (Figure 10) and when $N$ is smaller (Figure 11). In cases where $N$ exceeds the required dimensionality, sparse coding still strives to maintain orthogonal feature directions, leading to a near-identity matrix. However, both SAEs and MLPs show stronger correlations between features, as indicated by larger off-diagonal elements, though MLPs exhibit less extreme correlations (e.g., off-diagonal values of around 0.5).

When $N$ is smaller than the true sparse dimension (Figure 11), sparse coding again attempts to maintain orthogonality, though it is constrained by the reduced number of dimensions. The SAE

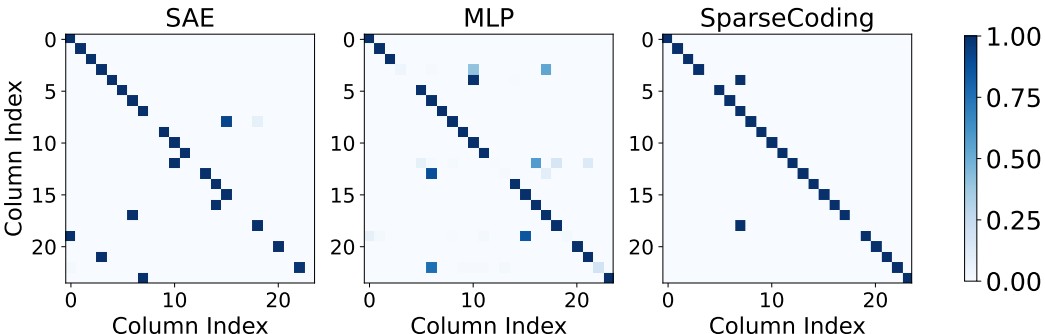

Figure 10: Visualisation of $D^\top D$ when $N$ exceeds the true sparse dimension.

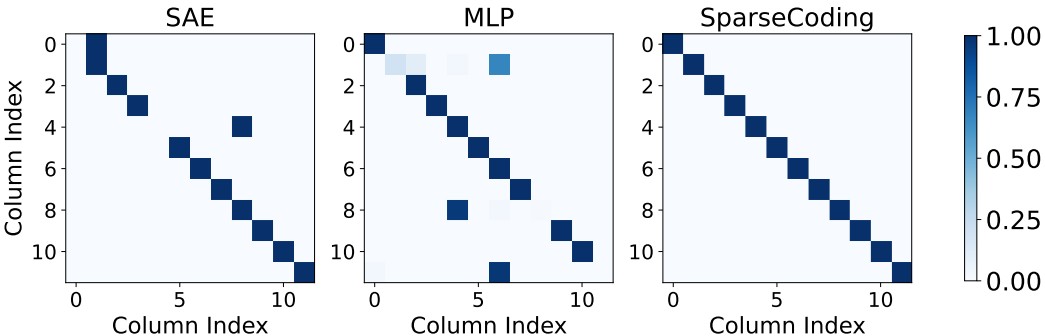

Figure 11: Visualisation of $D^\top D$ when $N$ is smaller than the true sparse dimension.

and MLP models, in contrast, continue to exhibit superposition, with off-diagonal elements close to 1.0. MLPs, however, show somewhat weaker correlations between features, as indicated by off-diagonal values around 0.5 in some instances.

## D    MLP ABLATIONS

We also wanted to understand in more fine-grained detail how the hidden width of the MLPs affects the key metrics of performance, in different regimes of $N, M$ and $K$. We show this in Figure 12. We use varying hidden widths and three different combinations of increasingly difficult $N, M, K$ to test this. We train for 50,000 iterations with a learning rate of 1e-4. We see that MCC (both latent and dictionary) increases approximately linearly with hidden width, with a slight drop-off at a hidden width of 512 (most likely due to underfitting). We also see a similar trend in terms of reconstruction loss, with the most difficult case being most sensitive to hidden width.

## E    INCLUDING A BIAS PARAMETER

We examine the effect of including a bias parameter in our models in Figure 13. Elhage et al. (2022) noted that a bias allows the model to set features it doesn't represent to their expected value. Further, ReLU in some cases can make "negative interference" (interference when a negative bias pushes activations below zero) between features free. Further, using a negative bias can convert small positive interferences into essentially being negative interferences, which helps deal with noise.

However, Theorem 1 doesn't rely on having biases, and although it generalises to the case with biases, we would like to be able to simplify our study by not including them. Thus, we show in Figure 13 that biases have no statistically significant effect on reconstruction loss, latent MCC, dictionary MCC, or L0, for any of the models, except for the L0 and MCC of the MLP, which achieves a higher MCC without bias at the cost of a greater L0.

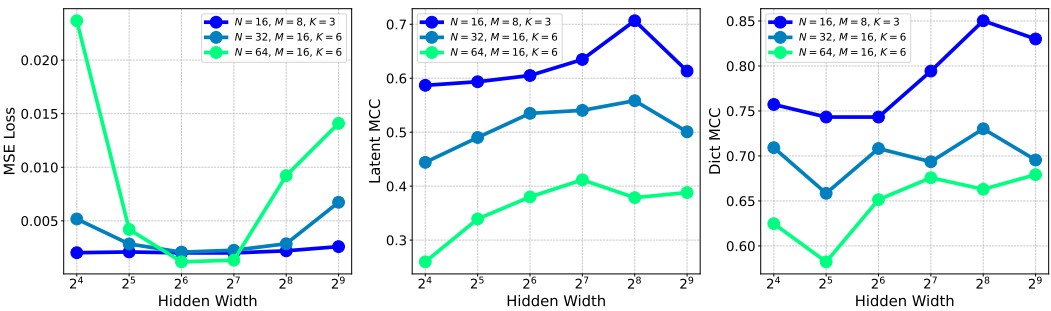

Figure 12: Varying the hidden width of an MLP autoencoder in varying difficulties of dictionary learning regimes. Each data point is an MLP trained for 50,000 iterations with a learning rate of 1e-4.

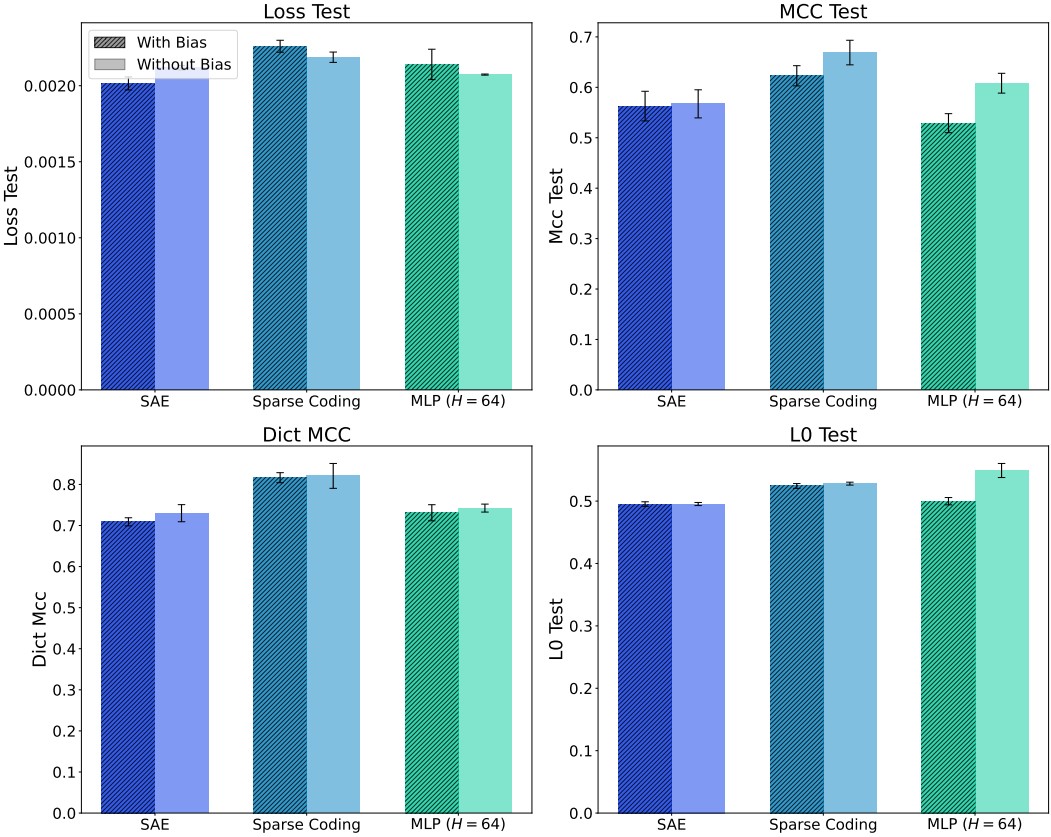

Figure 13: Effects on dictionary learning performance for our three models, with and without a bias. Including a bias has no statistically significant effect on results.

# F  COMPARISON WITH TRADITIONAL DICTIONARY LEARNING METHODS

To provide a comparison with traditional dictionary learning methods, we incorporated the Least Angle Regression (LARS) algorithm to compute the Lasso solution in our experimental framework.

The traditional dictionary learning problem can be formulated as a bi-level optimisation task. Given a set of training samples $X = [x_1, \ldots, x_n] \in \mathbb{R}^{m \times n}$, we aim to find a dictionary $D \in \mathbb{R}^{m \times k}$ and sparse codes $A = [\alpha_1, \ldots, \alpha_n] \in \mathbb{R}^{k \times n}$ that minimise the reconstruction error while enforcing sparsity constraints:

$$\min_{D,A} \sum_{i=1}^{n} \left( \frac{1}{2} \|x_i - D\alpha_i\|_2^2 + \lambda \|\alpha_i\|_1 \right)$$

subject to $\|d_j\|_2 \leq 1$ for $j = 1, \ldots, k$, where $d_j$ represents the $j$-th column of $D$, and $\lambda > 0$ is a regularisation parameter controlling the trade-off between reconstruction fidelity and sparsity.

In our experiment, we employed the LARS algorithm to solve the Lasso problem for sparse coding, while alternating with dictionary updates to learn the optimal dictionary. Specifically, we used the `scikit-learn` implementation of dictionary learning, which utilises LARS for the sparse coding step. The algorithm alternates between two main steps: (1) sparse coding, where LARS computes the Lasso solution for fixed $D$, and (2) dictionary update, where $D$ is optimised while keeping the sparse codes fixed.

To evaluate the performance of this traditional approach, we generated synthetic data following the same procedure as in our main experiments, with $N = 16$ sparse sources, $M = 8$ measurements, and $K = 3$ active components per timestep. We trained the dictionary learning model on the training set and evaluated its performance on the held-out test set. Performance was measured using the Mean Correlation Coefficient (MCC) between the predicted and true latents, as well as between the learned and true dictionary elements.

The results of this, presented in Figure 14, make clear that traditional sparse coding significantly outperforms our vanilla gradient-based implementations, particularly in terms of latent MCC and dictionary MCC. Whilst our results from the main body show that there does exist a significant amortisation gap between the vanilla implementations of each of the approaches, we should also attempt to understand how the optimised versions of each method compare. We discuss this in the following subsection.

## F.1  OPTIMISED SPARSE AUTOENCODERS AND SPARSE CODING

Our initial implementations of sparse autoencoders (SAEs) and sparse coding, while functional, are far from optimal. They represent the minimum computational mechanisms required to solve the problems as we have formulated them. However, more sophisticated approaches can significantly improve performance and address inherent limitations.

### F.1.1  ADVANCED SPARSE AUTOENCODER TECHNIQUES

Sparse autoencoders trained with L1 regularisation are susceptible to the *shrinkage problem*. Wright & Sharkey (2024) identified feature suppression in SAEs, analogous to the activation shrinkage first described by Tibshirani (1996) as a property of L1 penalties. The shrinkage problem occurs when L1 regularisation reduces the magnitude of non-zero coefficients to achieve a lower loss, potentially underestimating the true effect sizes of important features.

Several techniques have been proposed to mitigate this issue:

- **ProLU Activation**: Taggart (2024) introduced the ProLU activation function to maintain scale consistency in feature activations.

- **Gated SAEs**: Rajamanoharan et al. (2024a) developed Gated Sparse Autoencoders, which separate the process of determining active directions from estimating their magnitudes. This approach limits the undesirable side effects of L1 penalties and achieves a Pareto improvement over standard methods.

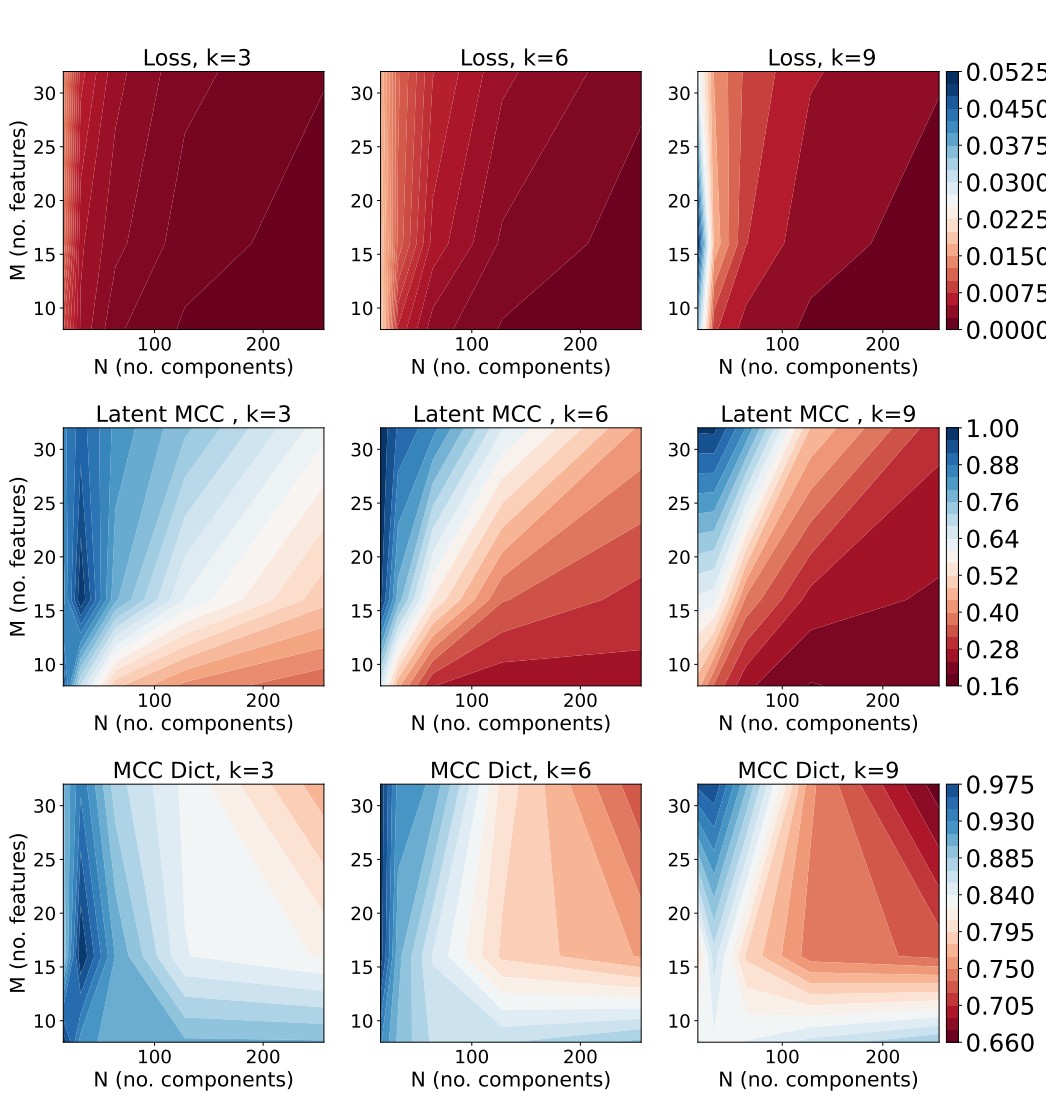

Figure 14: Performance of Least-Angle Regression (LARS) to compute the Lasso solution using our synthetic dictionary learning setup. In general, when comparing to Figure 5, we see an improvement when using LARS over our naïve implementations of SAEs, MLPs and sparse coding, across loss, latent MCC, and dictionary MCC.

- **JumpReLU SAEs**: Rajamanoharan et al. (2024b) proposed JumpReLU SAEs, which set activations below a certain threshold to zero, effectively creating a non-linear gating mechanism.
- **Top-k SAEs**: Originally proposed by Makhzani & Frey (2013), top-k SAEs were shown by Gao et al. (2024) to prevent activation shrinkage and scale effectively to large language models like GPT-4.

### F.1.2 Optimised Sparse Coding Approaches

Our initial sparse coding model, using uniformly initialised latents and concurrent gradient-based optimisation of both sparse codes and the dictionary, is suboptimal. The sparse coding literature offers several more sophisticated approaches:

- **Least Angle Regression (LARS)**: Introduced by Efron et al. (2004), LARS provides an efficient algorithm for computing the entire regularisation path of Lasso. It is particularly effective when the number of predictors is much larger than the number of observations.
- **Orthogonal Matching Pursuit (OMP)**: Pati et al. (1993) proposed OMP as a greedy algorithm that iteratively selects the dictionary element most correlated with the current residual. It offers a computationally efficient alternative to convex optimisation methods.

Future work will involve pitting these against the optimised SAE architectures discussed above.

### F.2 Top-$k$ sparse coding

Building on this exploration, we introduced a top-$k$ sparse coding approach. We aimed to determine whether (1) setting very small active latents to zero would improve performance and (2) optimising with a differentiable top-$k$ function, rather than using exponential or ReLU functions, could yield further benefits.

Figure 15 presents the results of these experiments. We first trained the sparse coding model for 20,000 steps on the training data and optimised for an additional 1,000 steps on the test data. During this process, we measured mean squared error (MSE) loss, latent MCC, and the $L_0$ norm of the latent codes. Due to the presence of very small active latents, all initial setups led to an $L_0$ value of 1.0, indicating that all latents were active, as shown by the blue star in the figure. We also show a sparse autoencoder trained with different $L_1$ penalties as a comparison.

Next, we applied a top-$k$ operation to enforce sparsity by setting all but the top-$k$ largest activations to zero. This process resulted in improved $L_0$ values, but the MSE loss and MCC results indicated that the top-$k$ optimisation itself was hampered by an insufficient learning rate. We hypothesise that with proper tuning of hyperparameters, we could achieve Pareto improvements by using the top-$k$ function directly, rather than applying it to exponentiated codes.

We believe that further adjustments to the optimisation process, including a higher learning rate for top-$k$ functions, could result in better performance. Additionally, applying the top-$k$ function directly, without exponentiating the codes, may offer further gains in performance and sparsity.

## G Measuring FLOPs

To quantify the computational cost of each method, we calculate the number of floating-point operations (FLOPs) required for both training and inference. This section details our approach to FLOP calculation for each method.

### G.1 Sparse Coding

For sparse coding, we calculate FLOPs for both inference and training separately.

**Inference:** The number of FLOPs for inference in sparse coding is given by:

$$\text{FLOPs}_{\text{SC-inf}} = \begin{cases} 3MN + Nn_s & \text{if learning } D \\ 2MN + Nn_s & \text{otherwise} \end{cases} \tag{17}$$

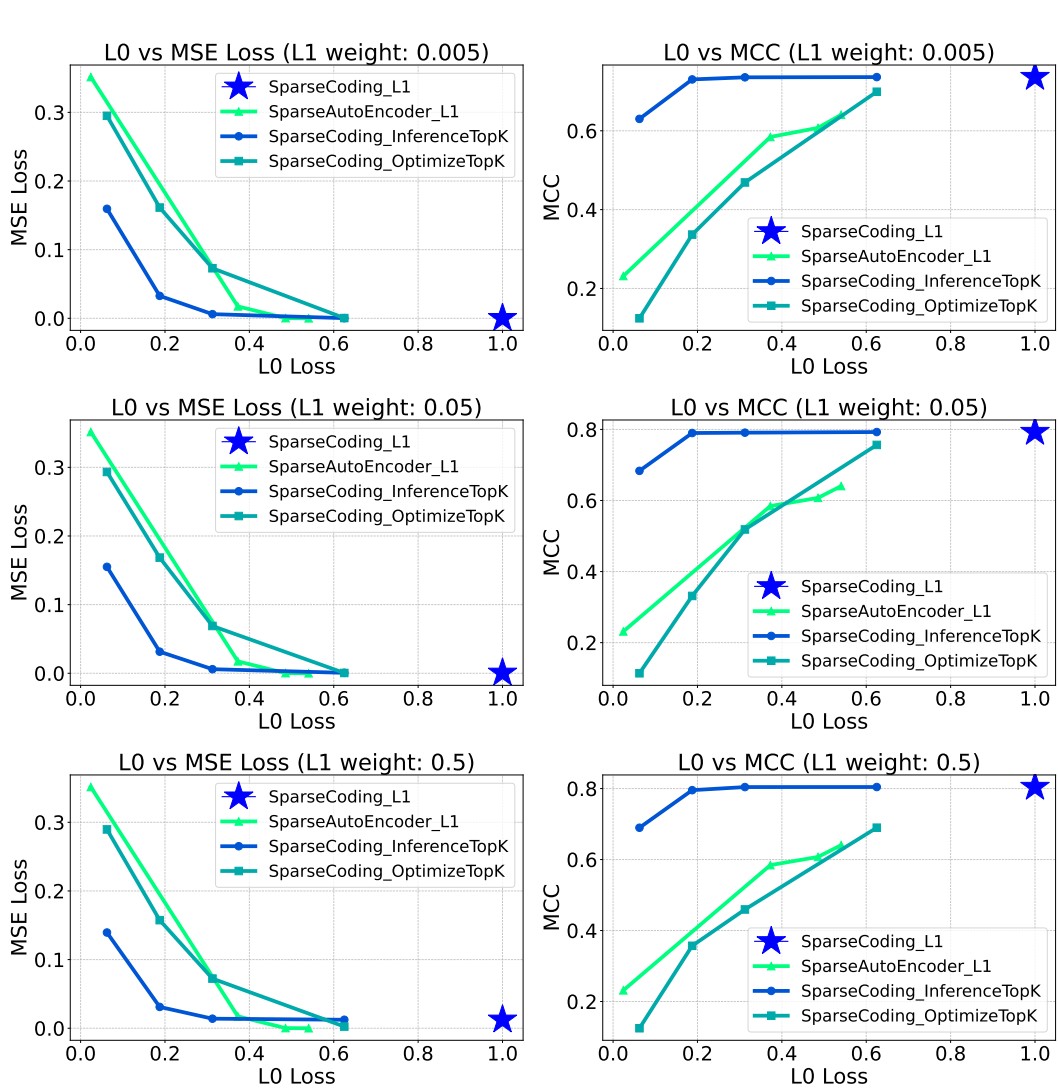

Figure 15: Comparison of $L_0$ loss vs. MSE loss and $L_0$ loss vs. MCC for Sparse Coding with L1 regularization, top-$k$ inference, and top-$k$ optimization, alongside results for Sparse Autoencoder. Blue stars represent the initial model's performance, while curves illustrate the results of applying top-$k$ sparsity.

where $M$ is the number of measurements, $N$ is the number of sparse sources, and $n_s$ is the number of samples. The additional $MN$ term when learning $D$ accounts for the normalisation of the dictionary.

**Training:** For training, we calculate the FLOPs as:

$$\text{FLOPs}_{\text{SC-train}} = n_{\text{eff}} \cdot (\text{FLOPs}_{\text{forward}} + \text{FLOPs}_{\text{loss}} + \text{FLOPs}_{\text{backward}} + \text{FLOPs}_{\text{update}}) \quad (18)$$

where $n_{\text{eff}} = n_{\text{steps}} \cdot \frac{n_b}{n_s}$ is the effective number of iterations, $n_{\text{steps}}$ is the number of training steps, $n_b$ is the batch size, and $n_s$ is the total number of samples. The component FLOPs are calculated as:

$$\text{FLOPs}_{\text{forward}} = \text{FLOPs}_{\text{SC-inf}} \quad (19)$$

$$\text{FLOPs}_{\text{loss}} = 2Mn_b + Nn_b \quad (20)$$

$$\text{FLOPs}_{\text{backward}} \approx 2 \cdot \text{FLOPs}_{\text{forward}} \quad (21)$$

$$\text{FLOPs}_{\text{update}} = \begin{cases} Nn_b + MN & \text{if learning } D \\ Nn_b & \text{otherwise} \end{cases} \quad (22)$$

## G.2 SPARSE AUTOENCODER (SAE)

For the sparse autoencoder, we calculate FLOPs for both training and inference.

**Training:** The total FLOPs for SAE training is given by:

$$\text{FLOPs}_{\text{SAE-train}} = n_{\text{eff}} \cdot (\text{FLOPs}_{\text{forward}} + \text{FLOPs}_{\text{backward}}) \quad (23)$$

where $n_{\text{eff}}$ is defined as before, and:

$$\text{FLOPs}_{\text{forward}} = \begin{cases} 5MN + N & \text{if learning } D \\ 4MN + N & \text{otherwise} \end{cases} \quad (24)$$

$$\text{FLOPs}_{\text{backward}} = N + (2NM + N) + 2NM + 2(MN + N) + \begin{cases} 2NM & \text{if learning } D \\ 0 & \text{otherwise} \end{cases} \quad (25)$$

**Inference:** For SAE inference, the FLOPs are calculated as:

$$\text{FLOPs}_{\text{SAE-inf}} = (4MN + N) \cdot n_s \quad (26)$$

## G.3 MULTILAYER PERCEPTRON (MLP)

For the MLP, we calculate FLOPs for both training and inference, considering a single hidden layer of size $H$.

**Training:** The total FLOPs for MLP training is given by:

$$\text{FLOPs}_{\text{MLP-train}} = n_{\text{eff}} \cdot (\text{FLOPs}_{\text{forward}} + \text{FLOPs}_{\text{backward}}) \quad (27)$$

where:

$$\text{FLOPs}_{\text{forward}} = \begin{cases} 2MH + H + 2HN + N + 2NM + MN & \text{if learning } D \\ 2MH + H + 2HN + N + 2NM & \text{otherwise} \end{cases} \quad (28)$$

$$\text{FLOPs}_{\text{backward}} = N + (2NH + N) + H + (2MH + H) + 2NM + 2(MH + H + HN + N) \quad (29)$$

where we add $2NM$ to $\text{FLOPs}_{\text{backward}}$ if learning $D$, and not otherwise.

**Inference:** For MLP inference, the FLOPs are calculated as:

$$\text{FLOPs}_{\text{MLP-inf}} = (2MH + H + 2HN + N + 2NM) \cdot n_s \quad (30)$$

## G.4 SAE WITH INFERENCE-TIME OPTIMISATION (SAE+ITO)

For SAE+ITO, we calculate the additional FLOPs required for optimizing the codes during inference:

$$\text{FLOPs}_{\text{ITO}} = (MN + N + n_{\text{iter}} \cdot (4MN + 2M + 11N)) \cdot n_s \quad (31)$$

where $n_{\text{iter}}$ is the number of optimisation iterations performed during inference.

These FLOP calculations provide a consistent measure of computational cost across all methods, allowing for fair comparisons of efficiency and performance trade-offs.

## H  AUTOMATED INTERPRETABILITY

In this section, we describe the automated interpretability pipeline used to understand and evaluate the features learned by sparse autoencoders (SAEs) and other models in the context of neuron activations within large language models (LLMs). The pipeline consists of two distinct tasks: feature interpretation and feature scoring. These tasks allow us to generate hypotheses about individual feature activations and to determine whether specific features are likely to activate given particular token contexts.

### H.1  FEATURE INTERPRETER PROMPT

We use a feature interpreter prompt to provide an explanation for a neuron's activation. The interpreter is tasked with analysing a neuron's behaviour, given both text examples and the logits predicted by the neuron. Below is a summary of how the interpreter prompt works:

*You are a meticulous AI researcher conducting an investigation into a specific neuron in a language model. Your goal is to provide an explanation that encapsulates the behavior of this neuron. You will be given a list of text examples on which the neuron activates. The specific tokens that cause the neuron to activate will appear between delimiters like* `<<this>>`*. If a sequence of consecutive tokens causes the neuron to activate, the entire sequence of tokens will be contained between delimiters* `<<just like this>>`*. Each example will also display the activation value in parentheses following the text. Your task is to produce a concise description of the neuron's behavior by describing the text features that activate it and suggesting what the neuron's role might be based on the tokens it predicts. If the text features or predicted tokens are uninformative, you can omit them from the explanation. The explanation should include an analysis of both the activating tokens and contextual patterns. You will be presented with tokens that the neuron boosts in the next token prediction, referred to as* `Top_logits`*, which may refine your understanding of the neuron's behavior. You should note the relationship between the tokens that activate the neuron and the tokens that appear in the* `Top_logits` *list. Your final response should provide a formatted explanation of what features of text cause the neuron to activate, written as:* `[EXPLANATION]: <your explanation>`*.*

### H.2  FEATURE SCORER PROMPT

After generating feature interpretations, we implemented a scoring prompt to predict whether a specific feature is likely to activate on a given token. This ensures that the explanations generated by the interpreter align with actual activations. The scoring prompt tasks the model with evaluating if the tokens marked in the examples are representative of the feature in question.

*You are provided with text examples where portions of the sentence strongly represent the feature, with these portions enclosed by* `<<` *and* `>>`*. Some of these examples might be mislabeled. Your job is to evaluate each example and return a binary response (1 if the tokens are correctly labeled, and 0 if they are mislabeled). The output must be a valid Python list with 1s and 0s, corresponding to the correct or incorrect labeling of each example.*

### H.3  EVALUATION OF AUTOMATED INTERPRETABILITY

To evaluate the accuracy of the interpretations generated by the feature interpreter and feature scorer, we compared model-generated explanations against held-out examples. The evaluation involved calculating the F1-score, which was done by presenting the model with a mix of correctly labeled and falsely labeled examples. The model was then tasked with predicting whether each token in the example represented a feature or not, based on the previously generated interpretation. By comparing the model's predictions with ground truth labels, we can assess how accurately the feature interpretation aligns with actual neuron activations. This process helps validate the interpretability of the features learned by SAEs, MLPs, and other models.

This pipeline is based on the work of Juang et al. (2024), which itself builds on the work of others. Bills et al. (2023) used GPT-4 to generate and simulate neuron explanations by analyzing text that strongly activated the neuron. Bricken et al. (2023) and Templeton (2024) applied similar techniques to analyze sparse autoencoder features. Templeton (2024) also introduced a specificity analysis to

rate explanations by using another LLM to predict activations based on the LLM-generated inter-
pretation. This provides a quantification of how interpretable a given neuron or feature actually is.
Gao et al. (2024) demonstrated that cheaper methods, such as Neuron to Graph (Foote et al.), which
uses $n$-gram based explanations, allow for a scalable feature labeling mechanism that does not rely
on expensive LLM computations.

