# OpenReview forum: "Compute Optimal Inference and Provable Amortisation Gap in Sparse Autoencoders"
_ICLR.cc/2025/Conference — ICLR 2025 Conference Withdrawn Submission_

### Official Review · Reviewer_BorF · 2024-11-01

**Soundness:** 4
**Presentation:** 3
**Contribution:** 2
**Rating:** 5
**Confidence:** 4

**Summary:**

The authors first theoretically show the existence of an “ammortization gap”: SAEs cannot perfectly reconstruct a sparse overcomplete basis in cases when sparse coding can. The authors then examine a suite of synthetic experiments that compare the performance of 1. naive SAEs, 2. SAEs with MLP encoders, 3. SAEs with an inference time optimized encoder, and 4. sparse coding. Most interestingly, sparse coding outperforms SAEs at large numbers of FLOPs. Finally, the authors train a naive SAE and an MLP SAE on the activations of a layer from GPT2, and show the the MLP SAE achieves a lower loss and contains similarly interpretable features.

**Strengths:**

- The authors apply sparse coding ideas to SAEs, something the mech interp field is probably not doing as much as it should be!
- The main theoretical result showing that existing SAEs cannot perfectly find any overcomplete sparse set of vectors is extremely interesting and relevant. To me, this is strong evidence that the worry over powerful encoders for SAEs is misguided. After all, our goal is to find the true features underlying model activations, even if the model itself would have trouble extracting a non-noisy version of these features. I think it would actually be extremely helpful if you put an argument like this in the introduction.
- The synthetic experiments are comprehensive and have interesting findings. The break down of the experiments into 3 different independent pieces is very helpful.
- MLP features are shown to be equally interpretable to SAE features

**Weaknesses:**

- The synthetic data experiments take up the majority of the paper, but these experiments have a couple large problems. The first problem is that the scale of the experiments is extremely small compared to actual SAEs, so it is hard to know how the synthetic results generalize. E.g. the following is a table containing my estimates for each quantity in reality vs. the experiments:

| Quantity           | Reality         | Experiments |
|--------------------|-----------------|-------------|
| N                  | 10k - 10M       | 16          |
| # training samples | 100M - 10B      | 1024        |
| K                  | 20 - 200         | 3 - 9       |
| M                  | 500 - 5000      | 8           |

-  Second, the distribution of latents used to generate the synthetic data is gaussian, but actual data is very much not so, see the plethora of recent work on structure in SAE latents (e.g. Not All Language Model Features Are Linear 2024, The Geometry of Categorical and Hierarchical Concepts in Large Language Models 2024). To be fair, the authors do note this second point in their limitations section.
- Experiments on GPT2 only compare SAEs and MLPs; why not SAE w/ ITO and sparse coding? Is it because sparse coding is computationally prohibitive? If so, this feels like a weakness that should be mentioned.
- The performance difference of the models in section 5 may be mostly due to different amount of training compute (MLPs are more expensive to train) or the different number of dead neurons. The authors acknowledge that a weakness of their work is not using SOTA SAE architectues like topk, jumprelu, or gated; since these architectures lead to less dead features, I suspect that it might erase most of the difference between the models. (There now exist very good libraries for easily training SOTA SAEs, e.g. SAELens or the Eleuther SAE library).
- The real model experiments are limited to one model, one layer, one SAE size, one sparsity, and one SAE architecutre. It would be much more helpful to vary at least 3 or 4 of these options.
- Figure 1 is confusing on first reading it, I didn’t really understand it until I read the proof. At least in mech interp, people don’t usually think of the “decoder” as producing activation space, but instead as the part of the SAE responsible for mapping estimated latents to the reconstruction. I think you should replace “decoder” with something like “linear down projection”.
- The theory shows SAEs cannot perfectly reconstruct sparse overcomplete bases, but maybe they almost can. Do you have some sense of how close an approximation with a linear projection followed by relu could be?

**Questions:**

- What deciles are the activating examples in section 5 from? Top activating examples are usually more interpretable, but do not provide a complete picture for how interpretable a feature is.
- Is the difference between feature interpretabilities actually statistically significant? If not, line 502 shouldn’t say “and in some cases exceeding”.
- Nit: is this the correct use of the word “amortized”? I am familiar with it from algorithm analysis, e.g. we can amortize the cost of an array resizing operation across calls to a function (I believe it is used similarly in everyday language), but I am not familiar with how it is used in this paper.
- I don’t understand how MCC (eqn 7) is applied to the alignment between learned dictionary vectors and true dictionary vectors. Could a metric like average nearest neighbor cosine sim make more sense?
- What are the “stepwise improvements in training” for SAE+ITO, line 362? I thought SAE+ITO didn’t train? . Are you running on normal SAE checkpoints at each part?

Overall, while I really like the ideas and structure of the paper, I think the paper would greatly benefit from stronger real model experiments as well as more realisitic synthetic experiments. The theoretical proof is the strongest part of the paper IMO, as it shows a potential fatal flaw of current SAEs, but on its own it is not enough for an accept rating because of the weaknesses described above.

---

### Official Review · Reviewer_M5Pt · 2024-11-02

**Soundness:** 2
**Presentation:** 3
**Contribution:** 2
**Rating:** 3
**Confidence:** 4

**Summary:**

This paper investigates the ability of Sparse Autoencoders to correctly recover from signals generated as sparse combinations of the columns of an incoherent, overcomplete dictionary. The authors claim that Sparse Autoencoders cannot recover the ground truth sparse code because the rank of the weight matrix in an SAE is bounded by the number of measurements (the dimension of the signal) which is less than the dimension of the sparse code. Having concluded that SAEs are insufficient, the authors explore alternative models for sparse code prediction such as MLPs and classical dictionary learning algorithms and show that they perform better in synthetic experiments. Finally the authors evaluate the interpretability of features learned by SAEs and MLPs using GPT-4 instances and show marginal improvement in the interpretability of MLP features.

**Strengths:**

The paper is written clearly and is easy to follow.

**Weaknesses:**

1. The theoretical argument in this paper seems narrow and does not necessarily support the conclusion that SAEs cannot learn sparse codes. The proof of theorem 1 argues that since the weight matrix of the encoder has rank at most $M$ (signal dimension/number of measurements), the encoder cannot recover all sparse codes since that would require a matrix of rank $N$ (sparse code dimension). This seems to contradict prior evidence from sparse recovery algorithms and work on autoencoders. If we apply this argument to a sparse coding algorithm like Iterative Soft Thresholding / Matching pursuit which also uses the low rank matrix $D^\top D$ (where $D$ is an overcomplete dictionary), we arrive at a contradiction since such sparse coding algorithms can provable recover sparse codes from even noisy measurements. Moreover in [1], the authors show (in theorem 3.1) that one layer of a ReLU autoencoder can recover the support of the sparse code with high probability if the weights are close to the ground truth dictionary $D$. Reference [2] shows that autoencoders trained to minimize reconstruction error with gradient descent can recover the ground truth dictionary $D$. The discrepancy between theorem 1 and these observations can possibly be explained by the fact that both sparse coding algorithms like ISTA and ReLU autoencoders use a nonlinearity between the encoder $W_e$ weights and the decoder weights $W_d$, and they do not try to predict the sparse code directly. Analyzing SAEs in this setting can resolve the apparent "amortization gap" identified by the authors.

2. Connections between sparse coding and autoencoders have been explored extensively in the literature. The application of deep networks to compressed sensing/sparse recovery problems has also been widely explored. This paper does not cite or engage with any of this literature.

[1] Rangamani, A., Mukherjee, A., Basu, A., Arora, A., Ganapathi, T., Chin, S., & Tran, T. D. (2018, June). Sparse coding and autoencoders. In 2018 IEEE International Symposium on Information Theory (ISIT) (pp. 36-40). IEEE.

[2] Nguyen, T. V., Wong, R. K., & Hegde, C. (2019, April). On the dynamics of gradient descent for autoencoders. In The 22nd International Conference on Artificial Intelligence and Statistics (pp. 2858-2867). PMLR.

[3] Arora, S., Ge, R., Ma, T., & Moitra, A. (2015, June). Simple, efficient, and neural algorithms for sparse coding. In Conference on learning theory (pp. 113-149). PMLR.

[4] Refinetti, M., & Goldt, S. (2022, June). The dynamics of representation learning in shallow, non-linear autoencoders. In International Conference on Machine Learning (pp. 18499-18519). PMLR.

[5] Kunin, D., Bloom, J., Goeva, A., & Seed, C. (2019, May). Loss landscapes of regularized linear autoencoders. In International conference on machine learning (pp. 3560-3569). PMLR.

[6] Zhang, Z., Liu, Y., Cao, X., Wen, F., & Zhu, C. (2021). Scalable deep compressive sensing. arXiv preprint arXiv:2101.08024. and references within for compressed sensing with deep networks.

[7] Gregor, K., & LeCun, Y. (2010, June). Learning fast approximations of sparse coding. In Proceedings of the 27th international conference on international conference on machine learning (pp. 399-406).

[8] Monga, V., Li, Y., & Eldar, Y. C. (2021). Algorithm unrolling: Interpretable, efficient deep learning for signal and image processing. IEEE Signal Processing Magazine, 38(2), 18-44.

3. The experiments seem to show that deeper MLPs perform better than shallow SAE architectures in sparse code prediction and dictionary learning problems. This is again intuitive and has been explored in the context of deep unfolding networks [7,8] and deep network models for sparse recovery (check [6] and references within). It is unclear what the new finding here is.

4. The interpretability experiments in section 5 could use more elaboration, specifically the part that describes evaluation with GPT-4. On the surface, the idea of evaluating the interpretability of LLM features using another LLM seems circular. Explaining the foundations behind this evaluation can aid the reader in understanding this section. The MLP results in this section also do not seem to be significantly better than the SAE results. Is this due to the evaluation technique, or a fundamental limitation of MLPs? The authors do not explore this.

**Questions:**

See weaknesses for most questions.

1. Why use the mean correlation coefficient over MSE for measuring the accuracy of sparse code recovery?
2. What is $d$ in the definition of MSE?

---

### Official Review · Reviewer_cPg9 · 2024-11-03

**Soundness:** 3
**Presentation:** 4
**Contribution:** 2
**Rating:** 6
**Confidence:** 3

**Summary:**

The paper builds on recent work studying sparse autoencoders (SAEs) for decoding neural representations. SAEs typically have a simple linear encoder + elementwise nonlinearity, but this paper studies the use of more powerful encoders. The authors prove that the current SAE encoder architecture can fail to recover sparse signals -- SAEs have a nonzero "amortization gap". The authors then conduct some empirical studies comparing different encoders which are more powerful than the typical SAE encoder. On synthetic data, they find that more powerful encoders can outperform the typical SAE encoder, even in a FLOP-matched setting (when the FLOP budget is high). The authors also train their more-powerful encoder models on GPT-2 activations, and find that the resulting latents are similarly interpretable to standard SAE latents.

**Strengths:**

* The paper is well-structured and well-written. The "Background and Related Work" section in particular is very thorough and clear.
* The (potentially more powerful) sparse encoding schemes that the authors experiment with are reasonable.
* The authors prove a theoretical result that may be of interest to the interpretability community about the limits of SAEs.
* Their experiments suggest that the interpretability community should probably experiment with more powerful encoders for SAEs
* The fact that their latents, using an MLP encoder, are similarly interpretable to standard SAE latents is an interesting result. It is unfortunately at quite a small scale (N=200 latents each), and it would be good to have a more systematic exploration of this topic, which I think is of practical interest to the interpretability community, and challenges an assumption that went into the original SAE design of Bricken et al. (2023).

**Weaknesses:**

* The experiments systematically comparing SAE performance with MLP encoders, SAE_ITO, etc. are done only on synthetic data which is generated with some strong assumptions that could not be representative of real LLM activation vectors. For instance, for the "Unknown sparse codes and dictionary" results, a dataset of 2048 samples was used, but real-world SAE training is typically done with hundreds of millions of samples in the single-epoch regime. There are limited experiments on real LLM activations.

**Questions:**

* Is the red line (SAE) missing from Figure 4c?
* Is it possible to run more experiments on real-world LLM activations or is that beyond your current computational budget?

---

### Note · Authors · 2024-11-22

**Comment:**

# Withdrawal of Paper #6137 and Response to Reviewers

We would like to withdraw our paper "Compute Optimal Inference and Provable Amortisation Gap in Sparse Autoencoders" from consideration at ICLR 2025. We appreciate the thorough and constructive feedback from all reviewers, which has helped shape our ongoing work in important ways.

## Current Extensions

We are actively addressing the key limitations identified in the reviews through several substantial extensions:
1. We are scaling up our experiments with GPT-2 activations to include:
   - A comparison of classical sparse coding approaches alongside the MLP and SAE implementations
   - Larger latent space dimensions to better match real-world applications
   - Multiple layers and model configurations
2. We are improving the synthetic experiments by:
   - Using Zipf-distributed latents to better approximate the hierarchical structure observed in real neural networks
   - Increasing the scale of training samples and feature dimensions to match realistic scenarios
   - Expanding the range of sparsity levels tested

## Acknowledgments
We sincerely thank the reviewers for their detailed feedback, particularly regarding:
- The need for more realistic experimental conditions
- The importance of connecting our work to existing literature on sparse coding and autoencoders
- The suggestion to examine the statistical significance of our interpretability results

These insights have been invaluable in strengthening our research direction and methodology. We look forward to incorporating these improvements into a future submission.

Sincerely,
The Authors

**Withdrawal Confirmation:**

I have read and agree with the venue's withdrawal policy on behalf of myself and my co-authors.